Early changes in the urine proteome in a rat liver tumour model

Zhang Yameng 1
Gao Yufei 2
Gao Youhe gaoyouhe@bnu.edu.cn 1
1 Department of Biochemistry and Molecular Biology, Beijing Normal University, Gene Engineering Drug and Biotechnology Beijing Key Laboratory , Beijing , China
2 College of Information Science and Technology, Beijing Normal University , Beijing , China
Demmers Jeroen
Electronic publication date: 2020 Feb 10
Publication date: 2020
Volume: 8
Electronic Location ID: e8462
Received 2019 Jun 21; Accepted 2019 Dec 26
Copyright: ©2020 Zhang et al.
Copyright year: 2020
Copyright holder: Zhang et al.
License: This is an open access article distributed under the terms of the Creative Commons Attribution License, which permits unrestricted use, distribution, reproduction and adaptation in any medium and for any purpose provided that it is properly attributed. For attribution, the original author(s), title, publication source (PeerJ) and either DOI or URL of the article must be cited.
License URL: https://creativecommons.org/licenses/by/4.0/

Keywords: Proteome, Urine, Biomarker, Liver tumour

Funding: National Key Research and Development Program of China 2018YFC0910202 2016 YFC 1306300 Beijing Natural Science Foundation 7172076 Beijing cooperative construction project 110651103 Beijing Normal University 11100704 Peking Union Medical College Hospital 2016-2.27 This work was supported by the National Key Research and Development Program of China (2018YFC0910202 and 2016 YFC 1306300), Beijing Natural Science Foundation (7172076), Beijing cooperative construction project (110651103), Beijing Normal University (11100704), and Peking Union Medical College Hospital (2016-2.27). The funders had no role in study design, data collection and analysis, decision to publish, or preparation of the manuscript.

==============================
Background

Urine, as a potential biomarker source among body fluids, can accumulate many early changes in the body due to the lack of mechanisms to maintain a homeostatic state. This study aims to detect early changes in the urinary proteome in a rat liver tumour model.

Methods

The tumour model was established with the Walker-256 carcinosarcoma cell line (W256). Urinary proteins at days 3, 5, 7 and 11 were profiled by liquid chromatography coupled with tandem mass spectrometry (LC-MS/MS). Compared with controls, differential proteins were selected. Associations of differential proteins with cancer were retrieved.

Results

At days 3, 5, 7 and 11, five, fifteen, eleven and twelve differential proteins were identified, respectively. Some of the differential proteins were reported to be associated with liver cancer. This differential urinary protein pattern was different from the patterns in W256 subcutaneous, lung metastasis and intracerebral tumour models.

Conclusions

This study demonstrates that (1) early changes in urinary proteins can be found in the rat liver tumour model; (2) urinary proteins can be used to differentiate the same tumour cells grown in different organs.

Introduction

Liver cancer is the third-ranking cause of cancer mortality in the world (Chen et al., 2017; Chiou & Lee, 2016). The early detection may prevent metastatic processes, which can significantly improve survival rates for cancer patients. Despite the technology to detect cancer has quickly advanced in the last decade, there are still many patients who cannot be diagnosed at early disease stages because of the heterogeneity of the clinical manifestations of this disease (Chen et al., 2011). To reduce the cancer mortality rate, novel approaches must be considered for early detection. One effective strategy to improve the prognosis of liver cancer is to find the tumour at the early stage when patients have no obvious symptoms, so that liver function can be preserved as much as possible and more effective treatments can be applied.

Currently, liver cancer diagnosis mainly relies on detection with imaging equipment (such as ultrasound, CT and MRI) and biomarkers. However, images are susceptible to operator experience, and it is difficult to distinguish between liver cancer and non-malignant hyperplasia. It can also be difficult to detect many small nodules at the early stage. Approximately 22% of early liver cancer imaging is not typical (Pahwa et al., 2014). On the other hand, tumour biomarkers are easier to be detected, but there are still many challenges for clinical applications. For instance, alpha-fetoprotein (AFP), which rapidly decreases in serum after birth and is maintained at a low level throughout adulthood, is the most widely used biomarker for liver cancer (Spangenberg, Thimme & Blum, 2006). However, serum AFP is not sufficient for diagnosing patients due to its poor sensitivity and specificity. Previous studies suggest that there is no single serum biomarker that can predict liver cancer with optimal sensitivity and specificity, especially at the early stage (Tsuchiya et al., 2015).

Urine can reflect many early changes in the body due to the lack of mechanisms to maintain a homeostatic state (Gao, 2013; Huang et al., 2015). Many studies have demonstrated that proteomic technology can be used to find potential biomarkers of different diseases in the urine, such as glomerular diseases (Wang et al., 2008), obstructive nephropathy (Yuan et al., 2015), hepatic fibrosis (Zhang et al., 2017), autoimmune myocarditis (Zhao et al., 2018), subcutaneous tumours (Wu, Guo & Gao, 2017) and glioma (Ni et al., 2018).

Animal model is a good tool for the studying disease urinary biomarkers, as the exact start of the disease is known and there is very few confounding factor.

This study aims to discover early urinary proteins changes in the W256 liver tumour model and investigate the ability of the urine proteome to differentiate the same tumour cells grown in different organs.

Materials & Methods

Animals

Male Wistar rats (130 ± 20 g) were purchased from Beijing Vital River Laboratory Animal Technology Co., Ltd. The animal license was SCXK (Beijing) 2016-0006. All experiments were approved by the Institutional Animal Care Use & Welfare Committee of the Institute of Basic Medical Sciences, Peking Union Medical College (Animal Welfare Assurance Number: ACUC-A02-2014-007). All rats were housed under a standard 12 h light/12 h dark cycle, and the room temperature and humidity were maintained at a standard level (22 ± 1 °C, 65–70%).

Experimental model establishment

A liver tumour model was established in this study. All Wistar rats were randomly divided into different groups: the control group (n = 7) and the Walker-256 tumour-bearing group (n = 12). Walker-256 (W256) carcinosarcoma cells were obtained from Cell Culture Center of Chinese Academy of Medical Sciences (Beijing, China). These cells were cultured in the ascitic fluid of Wistar rats. All cells were harvested from the rats who were given an intraperitoneal injection of 1 ×107 W256 cells after two cycles of 7 d cell passage. Then, W256 cells were resuspended in phosphate-buffered saline (PBS) before injection. The viability of the cells was detected by the Trypan blue exclusion test using a Neubauer chamber. After anaesthesia, the left medial lobe of the liver was exposed. W256 cells (2. 5 × 105) were visually injected under the hepatic capsule into this lobe. The injection volume was 0.1 ml. An equal volume of PBS was also injected into the same location in the control rats.

To monitor tumour progression, the livers of five experimental rats and two control rats were randomly harvested 3, 5, 7, and 11 d after injection. The more details about animals are shown in Table S1. At day 18, all rats were sacrificed, and their livers were histologically examined. For histopathology, the liver was fixed in formalin (4%) and embedded in paraffin. Then, all samples were sectioned and evaluated with haematoxylin and eosin (H&E) staining.

Urine sample preparation

Urine samples of three tumour-bearing rats and three control rats were collected at four time points: days 3, 5, 7 and 11. Without any treatment, urine was collected at least 6 ml from each rat by metabolic cage alone overnight. All rats were fasted while collecting the urine sample. The urine samples were centrifuged to remove impurities and fragments at 12,000 × g for 30 min at 4 °C and stored at −80 °C for later use. Before LC-MS/MS analysis, the 2 ml urine samples were thawed and transferred to centrifuge tubes for centrifugation at 12,000 × g for 30 min at 4 °C to remove impurities. The samples were mixed with three volumes of prechilled ethanol, and the supernatants were precipitated at −20 °C for 2 h. The mixtures were centrifuged for 30 min at 4 °C, the supernatant was removed, and the precipitate was dissolved in a configured lysis buffer (8 mol/L urea, 2 mol/L thiourea, 50 mmol/L Tris, and 25 mmol/L DTT). After the dissolution was completed, the centrifugation was continued at 12,000 × g for 30 min at 4 °C, and then the supernatant was preserved. The protein concentration was determined by the Bradford assay. The urinary proteins at different time points were digested using the FASP method (Wisniewski et al., 2009). One hundred micrograms of protein were added to the 10 kDa filter device (Pall, Port Washington, NY, USA) for each sample, and the protein was washed several times in sequence with a prepared urea buffer (UA, 8 mol/L urea, 0.1 mol/L Tris-HCl, pH 8.5) and 25 mmol/L NH4HCO3 solutions. The protein samples were reduced with 20 mmol/L dithiothreitol (DTT, Sigma) at 37 °C for 1 h and then added to 50 mmol/L iodoacetamide (IAA, Sigma) for 30 min in the dark. The samples were centrifuged at 14,000 × g for 30 min at 18 °C, washed with UA and NH4HCO3, and trypsin (enzyme-to-protein ratio of 1:50) was added to digest the samples overnight at 37 °C. Oasis HLB cartridges (Waters, Milford, MA) were used to desalt the peptide mixtures, dried by vacuum evaporation, and then labelled for storage at −80 °C.

LC-MS/MS analysis

An EASY-nLC 1200 HPLC system (Thermo Fisher Scientific, USA) was used to separate the peptides. First, the peptides were acidified with 0.1% formic acid, and their concentrations were determined by the BCA assay; the samples were then diluted to 0.5 µg/ µL with UA. Then, 1 µg of each peptide sample was loaded onto the trap column (Acclaim PepMap® 100, 75 µm ×100 mm, 2 µm, nanoViper C18) at 0.3 µL/min (column flow rate) for 1 h (elution time). The elution gradient of mobile phase B was 5% to 40% (mobile phase A: 0.1% formic acid; mobile phase B: 89.9% acetonitrile). A Thermo Orbitrap Fusion Lumos Tribrid mass spectrometer (Thermo Fisher Scientific, USA) was used for analysing the samples (Sun et al., 2005). Survey MS scans were acquired by the Orbitrap in a 350–1,550 m/z range with the resolution set to 120,000. For the MS/MS scan, the resolution was set at 30,000, and the HCD collision energy was 30. Dynamic exclusion was employed with a 30 s window. Fifteen urine samples from three experimental rats and three control rats at four time points (days 3, 5, 7, and 11) were chosen for MS analysis. For each sample, two technical replicate analyses were performed.

Data analysis

All MS data were searched using Mascot Daemon software (version 2.5.1, Matrix Science, UK) with the SwissProt_2017_02 database (taxonomy: Rattus; containing 7,992 sequences). The conditions included the following: trypsin digestion was selected, 2 sites of leaky cutting were allowed, cysteine was fixedly modified, methionine oxidation and protein N-terminal acetylation were mutagenic, peptide mass tolerance was set to 10 ppm, and fragment mass tolerance was set to 0.05 Da. One-way ANOVAs were performed for statistical analyses. Multiple comparisons were conducted using one-way ANOVA with the least significant difference (LSD) test or Bonferroni’s test. All the differential proteins were screened with the following criteria: proteins with at least two unique peptides were allowed; the fold change in increased group ≥1.5 and the fold change in decreased group ≤0.67; average spectral count of each rat in the high-abundance group ≥4. Group differences resulting in P <0.05 were identified as statistically significant. All results were expressed as the mean ± standard deviation.

Functional annotation of the differential proteins

All differential proteins identified at the different time points were analysed by DAVID 6.8 (https://david.ncifcrf.gov/) and Ingenuity Pathway Analysis (IPA, Mountain View, CA, USA) to determine the functional annotation. The proteins were described in detail according to four aspects: biological process, cellular component, molecular function and pathway.

Comparison methods of different W256 tumour models

The results of the W256 liver tumour model were compared with three different previously published studies from our laboratory: (1) W256 subcutaneous model (Wu, Guo & Gao, 2017); (2) W256 lung metastasis model (Wei et al., 2018); (3) W256 intracerebral tumour model (Zhang et al., 2018). In (1), the tumour-bearing rats were subcutaneously inoculated with 2 × 106 viable W256 cells in 200 µL of PBS into the right flank of the animal. In (2), the experimental group was injected with 2 ×106 viable W256 cells in 100 µL of PBS by tail-vein injection. In (3), the model was established as follows: five microliters of sterile normal saline containing 2000 W256 cells were injected into the brain using a 100 µL microsyringe. The experimental results reflect how changes in urinary proteins occur when the same tumour cells are grown in different organs.

The comparison details for (1), (2), and (3): 127, 139 and 102 differential urinary proteins were identified in these models, respectively; all the differential urinary proteins were compared with the W256 liver tumour model; the biological processes of these proteins were compared with the W256 liver tumour model at early stages (before the appearance of obvious pathology).

Results

Body weight and histopathological characterization over time

There was a significant difference in body weight between the tumour-bearing rats and the control rats at day 7 (Fig. 1). The average body weight of the tumour-bearing rats (n = 6) was lower than that of the controls (n = 5), and the reduction of food and water intake was observed in the tumour-bearing rats after W256 cell implantation. At day 16, a tumour-bearing rat died. All rats were sacrificed at day 18. H&E staining showed the pathological change after the tumour cells grown in the liver. At day 3, the H&E staining showed that there were no obvious pathological changes. At days 7 and 11, carcinosarcoma cells stained with H&E were observed under the microscope, and the liver tissues showed heterogeneously necrotizing tumours and liver tissue during tumour progression. At day 18, all the experimental rats exhibited tumour (Fig. 2).

Urine proteome changes in the W256 liver tumour model

To investigate how the urine proteome changes with tumour progression, urine samples of three experimental rats and three control rats were chosen for MS analysis at four time points (days 3, 5, 7, and 11). In total, 663 urinary proteins were identified as shown in Table S2. Among these proteins, there were 92 differential proteins, and only 83 differential proteins that had human orthologs changed significantly in all rats (fold change ≥1.5 or ≤0.67, P < 0.05; LSD test; Table S3). When using the Bonferroni’s test, there were 35 differential proteins that had human orthologs changed significantly (fold change ≥1.5 or ≤0.67, adjusted P < 0.05, Table 1 and Table S4). As can be shown in Table 1, at day 3, five differential proteins, one of which increased and four of which decreased, were identified. At day 5, fifteen differential proteins, four of which increased and eleven of which decreased, were identified. At day 7, eleven differential proteins, three of which increased and eight of which decreased, were identified. At day 11, twelve differential proteins, four of which increased and eight of which decreased, were identified. The unsupervised clustering analysis of all urinary proteins were shown in Fig. 3A. Xaa Pro dipeptidase (PEPD) and fructose-1,6-bisphosphatase 1 (F16P1) changed consistently at three time points (Fig. 3B).

Figure 1 Body weights of Walker 256 tumour-bearing rats.

The average body weight of the tumour group was significantly lower than that of the control group (n = 6 rats in the tumour group and n = 5 rats in the control group; * indicates P < 0.05; **indicates P < 0.01; ***indicates P < 0.001).

Figure 2 Histopathological characterization after injection with W256 cells (200X).

(A) H&E staining of the control rat. (B) H&E staining of the tumour-bearing rat at day 3. (C) H&E staining of the tumour-bearing rat at day 5. (D) H&E staining of the tumour-bearing rat at day 7. (E) H&E staining of the tumour-bearing rat at day 11. (F) H&E staining of the tumour-bearing rat at day 18.

At day 3, the only upregulated protein galectin-3-binding protein (LG3BP), is a secreted glycoprotein that has an affinity for galectins and extracellular matrix proteins, and LG3BP can also interact and regulate cell adhesion (Hellstern et al., 2002). It has been reported that LG3BP is considered as a poor prognosis biomarker in different types of malignancies (Grassadonia et al., 2002). Besides, LG3BP can be obtained from the urinary exosome (Pyong-Gon et al., 2011; Saraswat et al., 2015). It has been also reported to be associated with prostate cancer and acute rejection (Heger et al., 2015; Loftheim et al., 2012), and involved in the inflammatory response and tumour progression (Ferrari et al., 2019). Of the other unreported proteins, there were four downregulated proteins: lysosomal thioesterase (PPT2), Ig gamma-1 chain C region (IGHG1), vitamin D-binding protein (VTDB) and pro-cathepsin H (CATH).

At day 5, one (A1AG) of the upregulated proteins and four (ENOA, F16P1, PEPD and PRDX1) of the downregulated proteins were reported to be associated with liver cancer. Alpha-1-acid glycoprotein (A1AG) and Xaa Pro dipeptidase (PEPD) were reported to be a potential biomarker in HCC patients in serum samples (Ahn et al., 2012; Ilikhan et al., 2015). Alpha-enolase (ENOA) may serve as a candidate biomarker for early HCC diagnosis (Chen et al., 2010). Fructose-1, 6-bisphosphatase 1 (F16P1) and peroxiredoxin 1(PRDX1) were considered as potential biomarkers for the prognosis of liver cancer (Chen et al., 2016; Sun et al., 2015). In addition, the other unreported proteins (HA11, MTND, IC1, RNS1G, MDHC, MOES, NHRF1, IDHC, GSH1 and GSH0) may also have great potential to be used to predict liver cancer. According to their fold change values, some of them rank higher than the reported proteins. Although no relationship with liver cancer has been established, they may play important roles in other diseases, for example, moesin (MOES) differentially expressed in the breast cancer (Carmeci et al., 1998).

Table 1 Differential urinary proteins in W256 model.

Homo	Protein name	Day3	Day5	Day7	Day11	Reported to be related to liver cancer	Reported to be related to other diseases	
		Fold change	Adjusted P-value	Fold change	Adjusted P-value	Fold change	Adjusted P-value	Fold change	Adjusted P-value			
Q08380	Galectin-3-binding protein (LG3BP)	4.40	0.0006	–	–	–	–	–	–	–	–	
P01859	Ig gamma-1 chain C region (IGHG1)	0.65	0.0203	–	–	–	–	–	–	–	–	
Q9UMR5	Lysosomal thioesterase PPT2 (PPT2)	0.64	0.0394	–	–	–	–	–	–	–	–	
P02774	Vitamin D-binding protein (VTDB)	0.58	0.0047	–	–	–	–	0.41	0.0108	–	Lung cancer and colorectal cancer	
P09668	Pro-cathepsin H (CATH)	0.50	0.0299	–	–	–	–	–	–	–	–	
P01891	Class I histocompatibility antigen (HA11)	–	–	3.45	0.0120	–	–	3.29	0.0251	–	–	
Q9BV57	Androgen-responsive ARD-like protein 1 (MTND)	–	–	2.15	0.0299	–	–	–	–	–	–	
P05155	Plasma protease C1 inhibitor (IC1)	–	–	2.03	0.0240	–	–	–	–	–	Ovarian cancer	
P02763	Alpha-1-acid glycoprotein (A1AG)	–	–	1.58	0.0409	–	–	–	–	Serum	Bladder cancer and lung cancer	
P06733	Alpha-enolase (ENOA)	–	–	0.28	0.0249	–	–	–	–	Tissue	–	
P07998	Ribonuclease pancreatic gamma-type (RNS1G)	–	–	0.22	0.0319	–	–	–	–	–	–	
P09467	Fructose-1,6-bisphosphatase 1 (F16P1)	–	–	0.17	0.0215	0.20	0.0445	0.32	0.0467	Tissue	–	
P40925	Malate dehydrogenase (MDHC)	–	–	0.14	0.0278	–	–	–	–	–	–	
P26038	Moesin (MOES)	–	–	0.13	0.0417	–	–	–	–	–	Breast cancer	
O14745	Na(+)/H(+) exchange regulatory cofactor NHE-RF1 (NHRF1)	–	–	0.07	0.0285	–	–	–	–	–	–	
O75874	Isocitrate dehydrogenase [NADP] cytoplasmic (IDHC)	–	–	0.06	0.0485	–	–	–	–	–	–	
P48506	Glutamate–cysteine ligase catalytic subunit (GSH1)	–	–	0.06	0.0441	0.13	0.0464	–	–	–	–	
P12955	Xaa-Pro dipeptidase (PEPD)	–	–	0.05	0.0164	0.03	0.0203	0.03	0.0203	–	–	
P48507	Glutamate–cysteine ligase regulatory subunit (GSH0)	–	–	0.04	0.0059	–	–	–	–	–	–	
Q06830	Peroxiredoxin-1 (PRDX1)	–	–	0.03	0.0394	0.16	0.0408	–	–	Tissue	–	
P20472	Parvalbumin alpha (PRVA)	–	–	–	–	3.33	0.0352	–	–	–	–	
P19320	Vascular cell adhesion protein 1 (VCAM1)	–	–	–	–	2.58	0.0306	–	–	Serum	–	
P02748	Complement component C9 (CO9)	–	–	–	–	1.61	0.0485	–	–	–	Gastric cancer	
P22392	Nucleoside diphosphate kinase B (NDKB)	–	–	–	–	0.66	0.0394	–	–	–	–	
P00918	Carbonic anhydrase 2 (CAH2)	–	–	–	–	0.58	0.0203	–	–	–	–	
P01011	Serine protease inhibitor A3K (SPA3K)	–	–	–	–	0.48	0.0305	–	–	–	–	
Q08257	Quinone oxidoreductase (QOR)	–	–	–	–	0.13	0.0130	–	–	–	–	
P61769	Beta-2-microglobulin (B2MG)	2.58	0.0821	–	–	–	–	3.63	0.0043	Serum	–	
O95968	Prostatic steroid-binding protein C2 (PSC2)	–	–	–	–	–	–	1.82	0.0404	–	–	
Q9H008	Phospholysine phosphohistidine inorganic pyrophosphate phosphatase (LHPP)	–	–	–	–	–	–	1.77	0.0138	Serum	–	
P06396	Gelsolin (GELS)	–	–	–	–	–	–	0.61	0.0320	Tissue, serum	Cervical cancer, colorectal cancer and non-small cell lung cancer	
P27487	Dipeptidyl peptidase 4 (DPP4)	–	–	–	–	–	–	0.54	0.0102	–	–	
O75882	Attractin (ATRN)	–	–	–	–	–	–	0.51	0.0240	–	Malignant astrocytoma	
P05937	Calbindin (CALB1)	–	–	–	–	–	–	0.25	0.0100	–	Lung cancer	
Q13228	Methanethiol oxidase (SBP1)	–	–	–	–	–	–	0.05	0.0028	Tissue	–	
Notes.

- means does not reach the criteria (fold change ≥ 1.5 or ≤ 0.67 and adjust P-value <0.05) compared with control. The results of two parts (reported to be related to liver cancer and other diseases) were annotated from previous studies.

As the disease progressed, the number of proteins changed continuously at the last two time points (days 7 and 11). The pathological manifestations at these stages were also obvious. However, protein biomarker candidates were mainly selected at the two early time points, especially those proteins that changed continuously, such as VTDB, CATH, HA11, F16P1, GSH1, PEPD and PRDX1. Among all differential proteins, several proteins were not only associated with liver cancers but also differentially changed in other cancers, which indicates that it is difficult to distinguish cancer types only by one or two protein markers. It may be related to the mechanism of tumour development (Schreiber, Old & Smyth, 2011).

Obviously, the screening criteria which use Bonferroni’s test are more stringent with a lower false positive rate and a higher false negative rate. The results are more suitable for future validation and clinical application. However, the screening criteria which use LSD test are more relaxed with a higher false positive rate and a lower false negative rate. It can have more information and it is easier to find the correlation with biological functions. Besides, the screening criteria which use LSD test are more similar to the screening criteria of other W256 models and easier for comparison. Therefore, we provided both results of the two different screening criteria for different analysis: (1) the more stringent differential urinary proteins in Table 1 were used to discover the early urinary proteins changes in W256 liver tumour model; (2) the less stringent differential urinary proteins in Table S3 were used for the comparison of different tumour models and functional enrichment analysis.

Figure 3 Statistical analysis of the urine proteome of W256 liver tumour model.

(A) Hierarchical clustering of the 663 proteins from the 15 samples (twelve subjects in the tumour-bearing group and three in the control group) at four time points. Lines represent proteins, and the colors correlate with their abundance (red indicates more abundant, blueindicates less abundant). (B) The Venn diagram of 35 differential proteins identified at days 3, 5, 7 and 11.

Comparison of urinary proteins in different tumour models

The differential urinary proteins of four W256 tumour models (92 differential proteins in liver tumour model, 139 differential proteins in lung metastasis model, 102 differential proteins in intracerebral tumour model, and 127 differential proteins in subcutaneous model) at all time points were compared as shown in a Venn diagram (Fig. 4). The results indicate that urinary proteins patterns were different when the same tumour cells were grown in different organs. It can be seen from the Venn diagram that each model had a different number of unique differential urinary proteins. The 30, 48, 47, and 34 unique differential proteins were identified in the liver tumour model, the lung metastasis model, the intracerebral tumour model, and the subcutaneous model, respectively. Twenty-eight differential proteins had human orthologs were specially identified in the W256 liver tumour model compared with the other three models. The comparison procedure is presented in Fig. 5. Among the overlapping proteins of these four models, it can be found that (1) 14 proteins are detected in all models, among which 13 proteins have human orthologs. (2) Most of the overlapping proteins reappear in more than two models in different combinations. (3) Among the common proteins, 20 differential proteins have been reported to be associated with liver cancer, and some proteins have been identified as biomarkers in a variety of tumours.

Figure 4 The overlapping differential proteins in urine samples of the four different W256 tumour models.

The comparison data is from previously published studies (Wu, Guo & Gao, 2017; Wei et al., 2018; Zhang et al., 2018).

Among the 28 unique proteins of the liver tumour model, five proteins have been reported to be associated with liver cancer, including serum amyloid P component (SAMP), alpha-l-fucosidase (AFU), urokinase-type plasminogen activator (UROK), peroxiredoxin 6 (PRDX6), and peroxiredoxin 1 (PRDX1). SAMP and AFU are promising candidate biomarkers for HCC (Ferrín et al., 2014; Montaser, Sakr & Khalifa, 2012). UROK may be a potential therapeutic target of HCC (Atsushi, Minoru & Tohru, 2014). PRDX6 may be a candidate biomarker for early HCC diagnosis, and PRDX1 can predict poor prognosis for overall survival (Chen et al., 2010; Sun et al., 2015).

Figure 5 The comparison procedure of urinary proteins differentially expressed in the four models.

The comparison data is from previously published studies (Wu, Guo & Gao, 2017; Wei et al., 2018; Zhang et al., 2018).

The remaining 23 proteins have not been reported as biomarkers for liver cancer. However, some of these proteins were detectable at two time points, which may play important roles. For example, it has been reported that overexpression of cathepsin H (CATH) is related to several pathological states, including carcinoma and melanoma (Grassadonia et al., 2002). Protein AMBP, a liver-specific precursor, is also a precursor of the heme-binding protein that counteracts the disruption of free haemoglobin (Van den Berg et al., 2017). Cadherin-2 (CADH2) plays a role in the epithelial-to-mesenchymal transition, which is the process considered to contribute to carcinoma progression (Bram & Geert, 2013; Thiery et al., 2009). Endothelial cell-selective adhesion molecule (ESAM), a member of the immunoglobulin receptor family, mediates homophilic interactions between endothelial cells. ESAM has been suggested to have a special functional role in pathological angiogenic processes such as tumour growth (Tatsuro et al., 2003). Phosphoglycerate mutase 1 (PGAM1) is an important glycolytic enzyme that regulates many important biological processes, such as glycolysis, the pentose phosphate pathway and serine biosynthesis in cancer cells (Qu et al., 2017).

The comparisons show that the growth of tumours in different organs has both commonalities and individual differences. The urinary proteins have the potential to distinguish the same tumour cell grown in different organs.

Functional analysis of differential proteins

In the W256 liver tumour model, the functional analysis of differential proteins at days 3, 5, 7 and 11 was conducted by using DAVID, including categorizing the biological processes, cellular component, and molecular functions (Fig. 6). Ninety-two differential proteins (Table S3) were annotated. For biological processes, the innate immune response, retina homeostasis, response to drugs, negative regulation of endopeptidase activity, membrane-to-membrane docking, gluconeogenesis, complement activation classical pathway, and glycolytic process were significantly changed (Fig. 6A). At day 3, the innate immune response was the first respond to the tumour cells. At days 5 and 7, with the development of tumours in vivo, the glycolytic process, complement activation classical pathway, carbohydrate metabolic process, glutathione metabolic process, membrane-to-membrane docking, establishment of the endothelial barrier, and negative regulation of endopeptidase activity began to respond to the tumour changes. At day 11, the tumour grew further in the body. The carbohydrate metabolic process, innate immune response and oxidation–reduction process still responded to the tumour. For cellular component, most of the differential proteins were in the extracellular exosome, extracellular space, MHC class I protein complex, blood microparticle, and extracellular region. A little number of differential proteins come from organelles (Fig. 6B). For molecular function, endopeptidase inhibitor activity, identical protein binding, peroxiredoxin activity, and glutathione binding were overrepresented (Fig. 6C). These biological processes were associated with neoplastic progression. It can be further confirmed that the changes of urinary proteins were affected by the body’s response to the tumour cells. For the canonical pathway of Ingenuity Pathway Analysis, FXR/RXR activation, gluconeogenesis I, glycolysis I, LXR/RXR activation, acute phase response signalling, allograft rejection signalling, phagosome maturation, OX40 signalling pathway, Cdc42 signalling and NRF2-mediated oxidative stress response showed the obvious changes (Fig. 7).

Figure 6 Functional analysis of differential proteins at days 3, 5, 7 and 11 in W256 model.

(A) Biological process; (B) cellular component; (C) molecular function.

Figure 8 shows the comparison of biological processes of the liver tumour model with other models at early stages (Wei et al., 2018; Wu, Guo & Gao, 2017; Zhang et al., 2018). The urinary proteins of different W256 models reflect different biological processes, suggesting that the biological processes of the same tumour cell grown in different organs may be different. At the early stages of all the models, the biological processes are very different. In the liver tumour model, the biological processes mainly reflect an immune response and metabolism. This may be related to the function of liver, because liver is a central organ for homeostasis and carries out a wide range of functions, including metabolism, glycogen storage, drug detoxification, production of various serum proteins, and bile secretion (Atsushi, Minoru & Tohru, 2014). In the subcutaneous model, the biological processes are the primary response to various nutrients and ions. In the intracerebral tumour model, the recognition and migration of cells in biological processes are particularly significant. In the lung metastasis model, the biological processes include epithelial cell differentiation, regulation of immune system process, complement activation, classical pathway, ERK1 and ERK2 cascades, and inflammatory response. There is a large number of different early biological processes in the lung metastasis model. This could be due to the greater number of differential proteins at the early stage than those of the other three models.

Figure 7 IPA analysis of differential proteins at days 3, 5, 7 and 11 in W256 model.

Figure 8 The analysis of the early stages of biological processes in different W256 models.

(A) The liver tumour model; (B) the subcutaneous model; (C) the intracerebral tumour model; (D) the lung metastasis model. All the early biological processes are shown above. There are 34 early biological processes in the W256 lung metastasis model. For comparison, the same number of biological processes as the W256 liver tumour model was selected according to P-value. The comparison data is from previously published studies (Wu, Guo & Gao, 2017; Wei et al., 2018; Zhang et al., 2018).

Discussion

In this study, the urinary proteins changed significantly after the injection of W256 cells. The differential proteins were screened by fold change and the P-value both varied at different stages. Some proteins have been reported to be associated with liver cancer at the early stage. At days 3 and 5, six proteins (LG3BP, A1AG, ENOA, F16P1, PEPD and PRDX1) were related to liver cancer and other diseases (Table 1). Among other unreported proteins, the fold change of class I histocompatibility antigen (HA11) ranks the first in the significantly upregulated proteins at day 5. Although lack of the reports about these proteins associated with liver cancer or other liver diseases, these unreported proteins may still play an important role at the early stage of liver cancer. The panel combined with these proteins and other known biomarkers has the potential to diagnose the early stage diseases in the future. These findings in animal model may provide clues to detect the early changes in the urinary proteins for the clinical application. However, there are many confounding factors which can interfere the urinary proteins, such as food, drinks, environment and medications. It will be a great challenge to find the early changes associated with the disease.

The functional analysis demonstrates that LXR/RXR activation, acute-phase response signalling, production of nitric oxide and reactive oxygen species in macrophages, and the complement system were significantly enriched during tumour progression (Wu, Guo & Gao, 2017). Gluconeogenesis I and glycolysis I are also involved in the process of tumour development because of the increased glucose flux in tumour tissue, which is a common trait of human malignancies (Gambhir, 2002). Some common biological processes have been changed in the four W256 models, including acute-phase response signalling, LXR/RXR activation and the complement system.

Interestingly, compared with other W256 models, it can be found that urinary proteins patterns were different when the same tumour cells grown in different organs. The possible reason why the four models have 14 common proteins is that no matter which organ was injected with W256 tumour cells, it may cause the same reaction of the body, thereby lead to the same changes. It requires further studies to validate in the future. The different combinations of the common proteins are also important for diagnosis, because it is difficult to diagnose the type of tumour by using a single biomarker. The panel of biomarkers is more accurate and reliable. In addition, this study also confirms that urine can sensitively distinguish the changes of organs. It is essential to explore the potential of urinary biomarkers.

Conclusions

Urinary proteins change happens very early in the W256 liver tumour model. Some of the differential urinary proteins had been reported to be associated with liver cancer. Urinary proteins can be used to differentiate the same tumour cells grown in different organs. These findings may provide important information for the early diagnosis of liver cancer.

Supplemental Information

Table S1 The description of different treatments in rats

Click here for additional data file.

Table S2 The details of all the identified proteins in the W256 liver tumour model

Click here for additional data file.

Table S3 The details of differential proteins at different stages in the W256 liver tumour model

All the differential proteins were screened with the following criteria: proteins with at least two unique peptides were allowed; the fold change in increased group ≥ 1.5 and the fold change in decreased group ≤ 0.67; average spectral count of each rat in the high-abundance group ≥ 4. Group differences resulting in P < 0.05 were identified as statistically significant. Multiple comparisons were conducted using one-way ANOVA with the least significant difference (LSD) test.

Click here for additional data file.

Table S4 The details of 35 differential proteins

Click here for additional data file.

Additional Information and Declarations

Competing Interests

Author Contributions

Animal Ethics

Data Availability

The authors declare there are no competing interests.

Yameng Zhang conceived and designed the experiments, performed the experiments, analyzed the data, prepared figures and/or tables, authored or reviewed drafts of the paper, and approved the final draft.

Yufei Gao performed the experiments, prepared figures and/or tables, and approved the final draft.

Youhe Gao conceived and designed the experiments, authored or reviewed drafts of the paper, and approved the final draft.

The following information was supplied relating to ethical approvals (i.e., approving body and any reference numbers):

The Institutional Animal Care Use & Welfare Committee of the Institute of Basic Medical Sciences, Peking Union Medical College approved this research (ACUC-A02-2014-007).

The following information was supplied regarding data availability:

Data is available at Figshare: Zhang, Yameng (2020): The raw data of the article “Early changes in the urine proteome in a rat liver tumor model”. figshare. Dataset. https://doi.org/10.6084/m9.figshare.8293220.v1.

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
