# Peer review of "Early changes in the urine proteome in a rat liver tumour model"

_PeerJ, doi:10.7717/peerj.8462_

## Round 0.1 · original submission · Major Revisions

Based on the reviewers' comments, clarification is needed on the number of samples used in this study, missing time points, the description of the research question, the sample preparation and the statistical analysis of the data. In addition, issues about the English language were reported - these should be improved.

Reviewer 1 ·

Basic reporting

The English language should be improved throughout to ensure that an international audience can follow the rationale for the study, the Methods and the Results. Several sentences are difficult to follow and untangle. For example, the opening sentence of the subsection 'Urine collection and sample preparation' (lines 105-106) is ambiguous.

There have been a number of clinical and experimental publications to date in this expanding area of proteomics, It would be helpful to expand the literature review applied to clinical biomarkers of hepatocellular carcinoma (perhaps to include serum too) in the Introduction (lines 62-68). The authors might consider referencing the following papers:
https://www.ncbi.nlm.nih.gov/pubmed/27847095, https://www.ncbi.nlm.nih.gov/pubmed/26675302,
https://www.ncbi.nlm.nih.gov/pubmed/21518826.

It would be helpful for the reader to tabulate which animals were used for which study (histology, urine collection). It isn't clear why urine samples were collected from only three controls and three experimental animals, given the study started with seven controls and 12 experimental animals.

Given a range of papers have already shown early urinary biomarkers of HCC can be detected using proteomics, the hypothesis would benefit from being more specific. Whilst there is some value in another documentation study a more hypothesis-driven study would be appropriate.

Experimental design

This paper would benefit from a more specific Research Question.

Various subsections in The Methods are difficult to follow, including Tumor cell line and Culture, Rat models of liver cancer, Histological analysis, Urine collection and sample preparation.

Did the statistical tests allow for multiple comparisons? Whilst the replicates were used for mean+/-sd values, were the replicates also included in the independent sample t-tests? From Tables S1 and S2, were the controls measured at days 3,5,7 and 11 or only at baseline?

Validity of the findings

Which of the urinary proteomic findings may be attributed to changes in body weight?

The Results are discussed in detail, but as many of sentences in the Discussion are listing proteins it is difficult to follow. Early biomarkers should be the main focus of the Discussion, in line with the title of the paper.

Reviewer 2 ·

Basic reporting

The manuscript submitted by Zhang et al. is focused on early changes of urinary proteome in animal model of liver tumor. The subject is very important and proposal interesting. Although, everyone agrees with sentence at lines 343-344 (“…a panel of urinary differential proteins may be an ideal choice to improve the sensitivity and accuracy of early diagnosis for liver cancer”), however, they don’t allow important results; it is to underline some criticisms concerning both experimental diagram and results.

Experimental design

Experimental design needs to be improved, see specific comments.

Validity of the findings

Work don't report relevant results to hypotheses

Additional comments

The manuscript submitted by Zhang et al. is focused on early changes of urinary proteome in animal model of liver tumor. The subject is very important and proposal interesting. Although, everyone agrees with sentence at lines 343-344 (“…a panel of urinary differential proteins may be an ideal choice to improve the sensitivity and accuracy of early diagnosis for liver cancer”), however, they don’t allow important results; it is to underline some criticisms concerning both experimental diagram and results.
Authors used 7 controls and 12 tumor-bearing animals; but only 5 animals were used to collect liver and three of them for urine collection at different times.
Authors missed an important time-point; it should be interesting for proteome analysis.
Concerning sample preparation, urine required to be centrifuged after collection and before storage at -80°C to eliminate the urinary debris related to cells and membranes.
To improve validation of biomarkers it is mandatory to compare proteomes obtained from both urine and liver collected at the same time.
Because repeatability was not good within the time, it should be better to analyze a greater number of samples; in fact, authors report to have used 12 animals, but three of them seem to be used.
Concerning comparisons, authors have to perform kinetics (at each time-point including T0) for each animal to reduce variability due to specific animal. Important comparison is also treated versus control animals at the same time (day), because growth may be important for proteome changes.
Hierarchical clustering is not very good, because animals are not correctly segregated for time-points.
About sentence reported at lines 204-205, it is not possible to be in agreement because Figure 4 indicates that is limiting the organ where tumor cells were injected (only 19 proteins out of 275, resulted common); in fact, authors report at line 244-246 that “…the same tumor cell grown in different organs may be different.” It should be interesting to allow the identity of shared 19 proteins.
About discussed proteins, such as LGALS3BP, authors need to evaluate recent works concerning urinary proteome, specifically obtained from urinary exosome (see ExoCarta website and recent articles and reviews).
In the Introduction section authors report the importance of AFT as liver related tumor biomarker, but this protein was not identified in their analyses and no comments were done.
Authors collected interesting sets of samples, but they need to complete both proteome analysis (including livers and T0 time-points) and comparisons.
Minor Comments
It is not clear because authors made 10kDa filter (line 118).
Reference list is very long, but it is not appropriate.
Also, it is not clear what the treatments according the bioethical roles.
Collected volume of urine is not reported.

---

## Round 0.2 · Minor Revisions

The authors have replied to the questions and remarks that were proposed by the reviewers to a sufficient extent. The paper is in principle suitable for publication.

A few minor changes are needed: in the legends to Figs 4 and 5 there must be some mention to the provenance of the data for the other tumors, which are presently only mentioned in the Methods section. Also, the P-values in the final table should be subjected to a correction for multiple comparisons instead of shown "as is".

---

## Round 0.3 · accepted · Accept

All comments and questions were dealt with in a sufficient manner.